# Association between the ratio of serum uric acid to high density lipoprotein cholesterol and depressive symptoms in middle-aged and elderly Chinese

**Hongwei Zhao, Mingcheng Xu⊕, Yu Han, Shuang Liu\*⊕, Yongtai Gong\*⊕**

Department of Cardiology, The First Affiliated Hospital of Harbin Medical University, Harbin, China

⊕ These authors contributed equally to this work.
\* gongth@126.com (YG); 1164746228@qq.com (SL)

## Abstract

### Background

Previous studies have reported that metabolic syndrome is associated with depression. In recent years, the ratio of uric acid to high-density lipoprotein cholesterol (UHR) has been considered as a new effective marker of metabolic syndrome. The purpose of this study was to investigate the association between UHR and depression in people aged 45 years and older in China using the China Health and Retirement Longitudinal Study(CHARLS) database.

### Methods

A total of 10,396 subjects aged 45 years and above were included in this cross-sectional study. The Center for Epidemiologic Studies Depression Scale (CESD-10) was used to facilitate rapid screening and assessment of depression. A CESD-10 score of ≥ 10 was considered the critical value of depression. UHR was calculated from the ratio of serum uric acid (mg/dL) to high-density lipoprotein cholesterol (mg/dL). Linear regression and logistic regression were used to explore the relationship between UHR and depression, respectively. In addition, subgroup analysis and interaction tests were performed.

### Results

The study found that UHR was negatively associated with depression. In the fully adjusted model, every 1-unit increase in UHR was associated with a 14% lower odds of developing depression (OR = 0.14, 95% CI: 0.05-0.37). Participants in the highest quartile of UHR were 24% less likely to develop depression compared with participants in the lowest quartile (OR = 0.76, 95% CI: 0.67–0.87). The interaction analysis indicates that this negative correlation is more pronounced in the subgroup aged 60 years and above.

**Data availability statement:** Publicly available datasets were analyzed in this study. Data stored on figshare:10.6084/m9.figshare.28356035.

**Funding:** This study was supported by the National Natural Science Foundation of China (Grant No. 82000390).The funders had no role in study design, data collection and analysis, decision to publish, or preparation of the manuscript.

**Competing interests:** The authors have declared that no competing interests exist.

## Conclusion

UHR was significantly negatively correlated with depressive symptoms in the middle-aged and elderly Chinese population. However, further prospective studies are needed to accurately elucidate the causal relationship between increased UHR levels and the risk of depression. Therefore, larger cohort studies are needed to support these findings.

## 1. Introduction

Globally, the number of individuals diagnosed with depression increased by approximately 18% from 2005 to 2015 [1], and it is projected to become the leading cause of global disease burden by 2030 [2]. In China, more than 95 million people suffer from depressive symptoms, and this number is increasing year by year [3]. Depression not only leads to a significant decline in cognitive function and quality of life [4], but is also closely related to the occurrence or aggravation of serious diseases such as cardiovascular disease and diabetes, as well as a significant increase in suicide rates [5,6], thus causing a huge social burden. Therefore, it is particularly important to identify and effectively manage patients with depression.

Metabolic syndrome is a group of complex metabolic disorders, characterized by hypertension, dyslipidemia, central obesity, and insulin resistance, and is closely related to type 2 diabetes and increased mortality from cardiovascular disease [7,8]. Depression is one of the common mental disorders among the elderly. Japanese studies have shown a significant association between metabolic syndrome and depression [9]. Recently, the serum uric acid to high-density lipoprotein cholesterol ratio has been considered as a predictive marker of a novel metabolic syndrome [10–12]. At present, there are no studies reporting the relationship between the ratio of serum uric acid to high-density lipoprotein cholesterol and depression.

Therefore, we hypothesized that there may be a correlation between serum uric acid and HDL cholesterol ratio and depression. This study used the 2011 data of the China Health and Retirement Longitudinal Study (CHARLS) to study whether there is a correlation between the ratio of serum uric acid to high-density lipoprotein cholesterol and depression among middle-aged and elderly people in China, aiming to provide evidence for depression. Provide a new basis for prevention and control.

## 2. Methods

### 2.1. Study population

The data of this study are from the CHARLS public data in 2011. In 2011, the survey used a multi-stage stratified sampling method to select residents aged 45 years and above from 450 communities (villages) in 150 counties in 28 provinces (autonomous regions and municipalities) across the country. 2011 was the first survey, and a total of 17,708 subjects were collected in this survey. This study selected middle-aged and elderly people aged 45 years and above who completed SUA and HDL-C measurements as research subjects, excluding subjects with missing information on key variables such as general demographic indicators, behavioral lifestyle, SUA and HDL-C, and chronic disease history, and finally included 10,396 research subjects (Fig 1). The research protocol has been approved by the Biomedical Ethics Review Committee of Peking University (IRB00001052-11015), and all research participants have signed informed consent. For more information, please visit the official website of the CHARLS database: (http://charls.pku.edu.cn/).

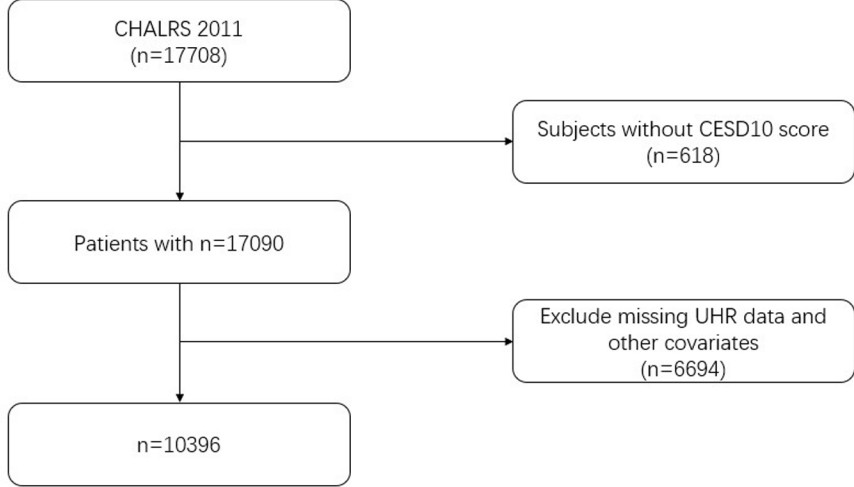

**Fig 1. Flowchart of subject selection.**

## 2.2. Definition of depressive symptoms

In the CHARLS database, the CES-D10 scale is used to assess the depressive symptoms of the subjects. This scale investigates the emotions and behaviors of the subjects in the past week and contains 10 question items, each of which is scored on a 4-point scale based on the frequency of occurrence. It should be noted that the scores of the 5th and 8th items need to be reversed. By adding up the scores of these items, the individual's CES-D10 total score can be calculated, ranging from 0 to 30 points. According to previous research designs [13–20], subjects with a total score of 10 or more are considered to have depressive symptoms, and the higher the score, the more severe the depressive symptoms.

## 2.3. Serum uric acid and high-density lipoprotein cholesterol

In the China Health and Retirement Longitudinal Study (CHARLS) project, professionally trained staff from the Chinese Center for Disease Control and Prevention (Chinese CDC) were responsible for collecting venous blood samples from participants. These blood samples were first quickly transported to the local laboratory at 4°C, followed by separation of plasma and buffy coat, and frozen at −20°C. After that, these samples were sent to the Chinese Center for Disease Control and Prevention in Beijing within two weeks and continued to be stored in a low-temperature environment of −80°C until they were sent to the Capital Medical University laboratory for relevant testing.

## 2.4. Other variables

This study extracted sociodemographic characteristics such as age, gender, place of residence, marital status, and education level, as well as health-related factors such as smoking status, drinking status, and chronic disease history from the questionnaire as possible confounding variables. Marital status was divided into married and unmarried (including separated, divorced, widowed, and never married). Education level was divided into below primary school, primary school, junior high school, high school, and above. Smoking was defined as whether or not smoked in the past, and drinking was defined as whether or not drank in the past. Chronic disease diagnoses were based on self-reports.

## 3. Statistical analysis

We chose quartiles as the cutoff points for the analysis because they are not normally distributed. This method can effectively stratify the sample and help us explore the nonlinear relationship and distribution characteristics of the variables. One-way analysis of variance was used to compare continuous variables among multiple groups, and categorical variables were described by frequency and percentage. The chi-square test was used to analyze the differences between different groups. Multivariate linear regression and multivariate logistic regression were used to explore the relationship between UHR and depression. In the partially adjusted model, we controlled for age and gender, and in the fully adjusted model, we included all covariates. In addition, we also performed subgroup analysis of the association between UHR and depression and tested their interactions. The purpose was to explore the mode of action of UHR in different subgroups of patients with depression and to evaluate the potential value of this biomarker in the prediction and diagnosis of depression. P values < 0.05 (two-sided) were considered statistically significant. All statistical analyses were performed using Empower software (www.empowerstats.com; X&Y solutions, Inc., Boston, MA) and R version 3.4.3 (http://www.R-project.org; The R Foundation).

## 4. Results

A total of 10,396 subjects were included in this study, with an average age of $(58.97 \pm 9.73)$ years. The subjects were divided into four groups according to the UHR quartiles: Q1 (<0.01), Q2 (0.01-0.07), Q3 (0.07-0.09) and Q4 (0.09-1.05). Compared with the lowest quartile, the depression scores of the remaining quartile groups were lower, and the proportion of depression was lower. In contrast, the proportion of women, smoking history, drinking history, hypertension, diabetes, dyslipidemia, heart disease, and stroke was higher. The baseline characteristics are shown in Table 1.

### Multivariate regression analysis of UHR with depressive score

Table 2 shows that lower UHR is significantly associated with higher depression scores, and this negative association remains significant in the fully adjusted model. For each unit increase in UHR, depression scores decreased by 4.5 points. (β = -4.50, 95% CI: -6.92, -2.08; p = 0.0003). In addition, multiple stratification variables (Q1-Q4) were also considered in the analysis, indicating that there are still significant differences in the relationship between UHR and depression scores in different groups. In the fully adjusted model, compared with Q1, the other quartile β values are -0.31, -0.52, and -0.68 respectively. These values indicate that for every unit increase in UHR, the depression score decreases by 0.31, 0.52, and 0.68 respectively.

### Multivariate regression analysis of UHR with depressive

The results showed that UHR was significantly negatively correlated with the risk of depression, that is, for every unit increase in UHR level, the risk of depression was reduced by 86%. After adjusting for other variables (Model I and Model II), although this negative relationship has weakened, it still maintains a significant difference (OR = 0.14, p < 0.0001). In addition, UHR stratified analysis results showed that compared with Q1, the risk of depression in the Q4 group was significantly reduced (OR = 0.76, p < 0.0001). For each unit increase in UHR, the risk of depression decreased by 24% (Table 3).

### Subgroup analyses

**UHR and depression scores.** Table 4 presents subgroup analysis based on age, gender, residence, marital status, and education level. Interaction tests revealed that, except for

**Table 1. Based on the demographic and clinical characteristics of UHR levels.**

| | Q1 (< 0.01) | Q2 (0.01-0.07) | Q3 (0.07-0.09) | Q4 (0.09-1.05) | p-value |
|---|---|---|---|---|---|
| **N** | 2599 | 2599 | 2599 | 2599 | |
| **Age (years)** | 58.30 ± 9.67 | 58.80 ± 10.01 | 59.34 ± 9.64 | 59.43 ± 9.56 | <0.001 |
| **Sex (N, %)** | | | | | <0.001 |
| Male | 1866 (71.80%) | 1549 (59.60%) | 1270 (48.86%) | 931 (35.82%) | |
| Female | 733 (28.20%) | 1050 (40.40%) | 1329 (51.14%) | 1668 (64.18%) | |
| **Education (N, %)** | | | | | <0.001 |
| Illiteracy | 1420 (54.64%) | 1287 (49.52%) | 1177 (45.29%) | 1008 (38.78%) | |
| Primary school | 519 (19.97%) | 568 (21.85%) | 590 (22.70%) | 591 (22.74%) | |
| Middle school | 455 (17.51%) | 497 (19.12%) | 541 (20.82%) | 617 (23.74%) | |
| High school or above | 205 (7.89%) | 247 (9.50%) | 291 (11.20%) | 383 (14.74%) | |
| **Married (N, %)** | 2242 (86.26%) | 2296 (88.34%) | 2309 (88.84%) | 2326 (89.50%) | 0.002 |
| **Residence (N, %)** | | | | | <0.001 |
| Rural | 742 (28.55%) | 905 (34.82%) | 983 (37.82%) | 1217 (46.83%) | |
| Urban | 1857 (71.45%) | 1694 (65.18%) | 1616 (62.18%) | 1382 (53.17%) | |
| **Smoking (N, %)** | 696 (26.78%) | 914 (35.17%) | 1099 (42.29%) | 1275 (49.06%) | <0.001 |
| **drinking (N, %)** | 807 (31.05%) | 903 (34.74%) | 1087 (41.82%) | 1198 (46.09%) | <0.001 |
| **Hypertension (N, %)** | 486 (18.70%) | 606 (23.32%) | 751 (28.90%) | 960 (36.94%) | <0.001 |
| **Diabetes (N, %)** | 115 (4.42%) | 133 (5.12%) | 186 (7.16%) | 214 (8.23%) | <0.001 |
| **Dyslipidemia (N, %)** | 150 (5.77%) | 223 (8.58%) | 274 (10.54%) | 430 (16.54%) | <0.001 |
| **Heart disease (N, %)** | 284 (10.93%) | 291 (11.20%) | 322 (12.39%) | 390 (15.01%) | <0.001 |
| **Stroke (N, %)** | 44 (1.69%) | 49 (1.89%) | 72 (2.77%) | 106 (4.08%) | <0.001 |
| **Cancer (N, %)** | 15 (0.58%) | 27 (1.04%) | 29 (1.12%) | 23 (0.88%) | 0.176 |
| **Lung disease (N, %)** | 247 (9.50%) | 280 (10.77%) | 237 (9.12%) | 236 (9.08%) | 0.131 |
| **Arthritis (N, %)** | 941 (36.21%) | 915 (35.21%) | 898 (34.55%) | 818 (31.47%) | 0.002 |
| **Liver disease (N, %)** | 91 (3.50%) | 72 (2.77%) | 91 (3.50%) | 102 (3.92%) | 0.143 |
| **Kidney disease (N, %)** | 147 (5.66%) | 152 (5.85%) | 153 (5.89%) | 157 (6.04%) | 0.950 |
| **Gastropathy (N, %)** | 683 (26.28%) | 604 (23.24%) | 564 (21.70%) | 454 (17.47%) | <0.001 |
| **UHR** | 0.05 ± 0.01 | 0.08 ± 0.01 | 0.10 ± 0.01 | 0.16 ± 0.05 | <0.001 |
| **CESD-10 Score** | 9.35 ± 6.63 | 8.67 ± 6.29 | 8.25 ± 6.17 | 7.69 ± 6.16 | <0.001 |
| **Depressive (N, %)** | 1127 (43.36%) | 1025 (39.44%) | 941 (36.21%) | 817 (31.44%) | <0.001 |

**Table 2. Association between UHR and depressive score.**

| | Unadjusted β (95% CI) p-value | Model I β (95% CI) p-value | Model II β (95% CI) p-value |
|---|---|---|---|
| **UHR** | −10.88 (−13.34, −8.42) < 0.0001 | −6.62 (−9.11, −4.12) < 0.0001 | −4.50 (−6.92, −2.08) 0.0003 |
| **Categories** | | | |
| Q1 | 0 | 0 | 0 |
| Q2 | −0.68 (−1.02, −0.34) 0.0001 | −0.49 (−0.83, −0.15) 0.0049 | −0.31 (−0.63, 0.01) 0.0566 |
| Q3 | −1.10 (−1.45, −0.76) < 0.0001 | −0.75 (−1.09, −0.40) < 0.0001 | −0.52 (−0.84, −0.19) 0.0019 |
| Q4 | −1.66 (−2.01, −1.32) < 0.0001 | −1.07 (−1.42, −0.72) < 0.0001 | −0.68 (−1.02, −0.34) < 0.0001 |

age, there were no significant interactions between UHR and depression scores across the different subgroups (all P > 0.05). This indicates that gender, residence, marital status, and education level do not significantly influence the relationship between UHR and depression scores.

**Table 3. Association between UHR and depressive.**

|  | Unadjusted OR (95% CI) p-value | Model I OR (95% CI) p-value | Model II OR (95% CI) p-value |
|---|---|---|---|
| **UHR** | 0.02 (0.01, 0.06) < 0.0001 | 0.09 (0.04, 0.22) < 0.0001 | 0.14 (0.05, 0.37) < 0.0001 |
| **Categories** |  |  |  |
| Q1 | 1.0 | 1.0 | 1.0 |
| Q2 | 0.85 (0.76, 0.95) 0.0041 | 0.90 (0.80, 1.00) 0.0606 | 0.94 (0.83, 1.06) 0.2902 |
| Q3 | 0.74 (0.66, 0.83) < 0.0001 | 0.82 (0.73, 0.92) 0.0006 | 0.86 (0.76, 0.97) 0.0127 |
| Q4 | 0.60 (0.53, 0.67) < 0.0001 | 0.71 (0.63, 0.80) < 0.0001 | 0.76 (0.67, 0.87) < 0.0001 |

**Table 4. Subgroup analyses of UHR and depression scores.**

|  | N | β (95% CI) | p-value | p for interaction |
|---|---|---|---|---|
| **Age** |  |  |  | 0.0071 |
| <60 | 5789 | −1.66 (−4.82, 1.50) | 0.3026 |  |
| ≥60 | 4607 | −8.12 (−11.73, −4.51) | <0.0001 |  |
| **Gender** |  |  |  | 0.2435 |
| Male | 5616 | −6.21 (−9.96, −2.46) | 0.0012 |  |
| Female | 4780 | −3.36 (−6.45, −0.27) | 0.0333 |  |
| **Residence** |  |  |  | 0.5304 |
| Rural | 6549 | −3.92 (−7.09, −0.75) | 0.0154 |  |
| Urban | 3847 | −5.47 (−9.18, −1.76) | 0.0038 |  |
| **Married** |  |  |  | 0.7164 |
| Yes | 9173 | −4.51 (−7.11, −1.90) | 0.0007 |  |
| No | 1223 | −3.21 (−9.72, 3.29) | 0.3330 |  |
| **Education** |  |  |  | 0.7729 |
| Illiteracy | 4892 | −5.89 (−9.67, −2.12) | 0.0022 |  |
| Primary school | 2268 | −4.33 (−8.96, 0.30) | 0.0671 |  |
| Middle school | 2110 | −4.02 (−9.37, 1.32) | 0.1403 |  |
| High school or above | 1126 | −1.88 (−8.79, 5.03) | 0.5944 |  |

**UHR and depression risk.** The results of the depression risk subgroup analysis are generally consistent with those of the depression score subgroup analysis. Except for age, no significant effects of other subgroups on the relationship between UHR and depression scores were found (p > 0.05) (Table 5).

## 5. Discussion

This study is a cross-sectional analysis based on the CHARLS database, which included 10,396 participants. To validate the stability of the results, we conducted both linear regression and logistic regression models, adjusting for confounding factors to derive more reliable conclusions. The findings revealed a significant association between lower UHR levels and higher depression risk. As UHR levels increased, both depression scores and the proportion of individuals with depression decreased. After adjusting for confounders, multiple linear regression and logistic regression analyses confirmed a significant negative correlation between UHR and depression scores, as well as depression risk. Moreover, subgroup analysis and interaction tests indicated a stronger association between UHR levels and depression status in individuals aged 60 and above. Older adults often face various stressors, such as physical disabilities, loss

**Table 5. Subgroup analyses of UHR and depression risk.**

|  | N | OR (95% CI) | p-value | p for interaction |
|---|---|---|---|---|
| **Age** |  |  |  | 0.0420 |
| <60 | 5789 | 0.34 (0.10, 1.20) | 0.0943 |  |
| ≥60 | 4607 | 0.05 (0.01, 0.20) | <0.0001 |  |
| **Gender** |  |  |  | 0.5589 |
| Male | 5616 | 0.10 (0.02, 0.43) | 0.0019 |  |
| Female | 4780 | 0.18 (0.05, 0.66) | 0.0095 |  |
| **Residence** |  |  |  | 0.3805 |
| Rural | 6549 | 0.20 (0.06, 0.67) | 0.0092 |  |
| Urban | 3847 | 0.08 (0.02, 0.39) | 0.0015 |  |
| **Married** |  |  |  | 0.1457 |
| Yes | 9173 | 0.11 (0.04, 0.31) | <0.0001 |  |
| No | 1223 | 0.77 (0.07, 8.17) | 0.8307 |  |
| **Education** |  |  |  | 0.8866 |
| Illiteracy | 4892 | 0.16 (0.04, 0.67) | 0.0117 |  |
| Primary school | 2268 | 0.10 (0.01, 0.74) | 0.0244 |  |
| Middle school | 2110 | 0.09 (0.01, 0.82) | 0.0331 |  |
| High school or above | 1126 | 0.33 (0.02, 6.84) | 0.4700 |  |

of loved ones, cognitive decline, and social isolation, which may contribute to the increased incidence and prevalence of depression with age [21]. These findings were consistent with the interaction effects observed in both linear and logistic regression analyses.

The role of UHR in depression remains underexplored, although the relationship between UA, HDL-C, and depression has been widely discussed. The mechanisms underlying the negative correlation between serum uric acid (UA) levels and the risk of depressive symptoms in adults can be explained from several perspectives. First, UA, as an endogenous antioxidant, clears free radicals and reactive oxygen species (ROS), alleviating oxidative stress and preventing neuronal damage [22]. Second, UA exerts neuroprotective effects by reducing neuroinflammation and preventing neuronal apoptosis, thereby maintaining neural function [23]. Additionally, UA has anti-inflammatory properties, as it can suppress pro-inflammatory cytokine production, thus reducing the inflammation associated with depression [24]. In addition, UA can modulate neurotransmitter systems by enhancing dopamine release and receptor function, as well as regulating glutamate transmission, both of which are closely linked to the pathophysiology of depression [25]. In summary, UA lowers depression risk through mechanisms such as antioxidant activity, neuroprotection, and anti-inflammation. A study from the NHANES database found a significant nonlinear negative correlation between serum UA levels and depressive symptoms in U.S. adults, with a turning point at 319.72 μmol/L [26]. Another study conducted in China found a negative correlation between serum uric acid and depression risk in East Asian populations [27]. Moreover, HDL-C has been shown to induce anti-inflammatory effects, with low HDL-C levels triggering significant inflammatory responses, which in turn increase the risk of depression [28–31]. A meta-analysis reported an association between lower HDL-C levels and depressive symptoms [32]. A study from Korea also found a negative correlation between HDL-C levels and depression [33]. Furthermore, another study examining the neurobiological links between depression and HDL-C level changes further confirmed the association between the two [34]. A study from Finland found that low HDL-C levels were associated with major depressive disorder in participants with a 7-year history of depressive symptoms [28]. A logistic regression analysis indicated that low

HDL-C levels were linked to longer symptom duration in patients with major depressive disorder [35]. By combining these two biomarkers, UHR offers a more comprehensive reflection of the interplay between oxidative stress and anti-inflammatory processes. In conclusion, our study provides further evidence using the novel inflammatory metabolic parameter UHR, enhancing the understanding of the relationship between UHR and depression risk.

This study has several significant strengths. First, based on existing data, it is the first to explore the association between UHR and depressive symptoms in middle-aged and elderly individuals in China. Second, UHR, as an easily accessible biomarker, provides valuable support for investigating its relationship with depressive symptoms. Lastly, the large sample size enhances the statistical power and generalizability of the results, offering meaningful insights into the potential role of UHR as a biomarker for depression in middle-aged and elderly populations, further elucidating the mechanistic link between metabolic dysregulation and mental health outcomes.

However, several limitations should be acknowledged. First, as a cross-sectional study, it cannot establish causal relationships between UHR and depression. Second, although the CESD-10 is useful for screening, it is subject to self-report biases, reduced diagnostic sensitivity, potential cultural differences, and limitations in detecting mild or subclinical depression. Additionally, participants with missing data on key variables were excluded from the analysis. This exclusion may introduce selection bias, as missing data may be related to factors such as the severity of depressive symptoms, sociodemographic characteristics, or health conditions. As a result, the findings may not fully represent the target population, potentially affecting the generalizability of the results. Despite rigorous data cleaning and preprocessing, future studies may consider using more refined imputation methods (e.g., multiple imputation) to mitigate the impact of missing data on the results and enhance the external validity of the findings. Finally, due to the lack of relevant data in the CHARLS database, certain potential confounding factors, such as dietary habits or physical activity, were not adjusted for in the models. These factors should be considered in future research.

## 6. Conclusion

The study indicates a significant association between UHR and the increased risk of depression in the Chinese middle-aged and elderly population. However, to accurately establish the causal relationship between elevated UHR levels and depression risk, further prospective studies are required. Larger cohort studies are necessary to substantiate these findings.

## Author contributions

**Conceptualization:** Mingcheng Xu, Yu Han, Yongtai Gong.

**Data curation:** Hongwei Zhao, Mingcheng Xu, Yu Han, Shuang Liu, Yongtai Gong.

**Formal analysis:** Hongwei Zhao, Yu Han, Shuang Liu.

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
