## [Decision Letter · Decision Letter 0]

15 Jan 2025

PONE-D-24-56380Relationship between the ratio of serum uric acid to high density lipoprotein cholesterol and depressive symptoms in middle-aged and elderly ChinesePLOS ONE

Dear Dr. xu,

Thank you for submitting your manuscript to PLOS ONE. After careful consideration, we feel that it has merit but does not fully meet PLOS ONE’s publication criteria as it currently stands. Therefore, we invite you to submit a revised version of the manuscript that addresses the points raised during the review process.

We look forward to receiving your revised manuscript.

Kind regards,

Vinod Kumar Yata, PhD

Academic Editor

PLOS ONE

https://www.frontiersin.org/journals/aging-neuroscience/articles/10.3389/fnagi.2022.919430/full

In your revision ensure you cite all your sources (including your own works), and quote or rephrase any duplicated text outside the methods section. Further consideration is dependent on these concerns being addressed.

“This study was supported by the National Natural Science Foundation of China (Grant No. 82000390) and the Youth Science Foundation Project (Grant No. 81100121).”

Reviewers' comments:

Reviewer's Responses to Questions

**Comments to the Author**

1. Is the manuscript technically sound, and do the data support the conclusions?

Reviewer #1: Yes

Reviewer #2: Yes

2. Has the statistical analysis been performed appropriately and rigorously? 

Reviewer #1: Yes

Reviewer #2: Yes

3. Have the authors made all data underlying the findings in their manuscript fully available?

Reviewer #1: Yes

Reviewer #2: Yes

4. Is the manuscript presented in an intelligible fashion and written in standard English?

Reviewer #1: No

Reviewer #2: Yes

5. Review Comments to the Author

Reviewer #1: Upon reviewing this manuscript, a primary concern arises, regarding the scientific value of using UHR as a marker for depression, given its primary establishment in previous studies as a predictor of metabolic syndrome (MS). This raises a fundamental "chicken-and-egg" dilemma: MS has been strongly implicated as both a potential cause and a consequence of depression, owing to shared mechanisms such as chronic inflammation, oxidative stress, and insulin resistance. Without addressing this bidirectional relationship, the rationale for UHR serving as an independent marker of depression remains unclear. It would strengthen the manuscript to discuss how UHR might uniquely capture depression-related metabolic dysregulation beyond its established role in MS prediction.

Nevertheless, this manuscript provides a valuable contribution to the field by exploring the novel relationship between UHR and depression, particularly leveraging a robust dataset of over 10,000 participants (However, this is 2011 CHARLS database, which might not entirely reflect current trends or health profiles). The large sample size enhances the statistical power and generalizability of the findings, offering meaningful insights into the potential role of UHR as a biomarker for depression in middle-aged and elderly populations. This study also lays a foundation for future research to elucidate further the mechanistic links between metabolic dysregulation and mental health outcomes.

This manuscript requires substantial revisions to address several critical deficiencies if it is to be accepted. Among the most significant issues are the following (please also refer to comments in the manuscript (PDF):

1. The cross-sectional design limits the ability to infer causal relationships between UHR and depressive symptoms. This limitation is acknowledged but could benefit from further elaboration on its implications and the necessity of longitudinal studies.

2. While many confounders were adjusted for, there might be unmeasured or residual confounders that influence the results, such as dietary habits or physical activity.

3. The manuscript mentions excluding subjects with missing key variable data. This might lead to selection bias, which should be addressed in the discussion.

4. While the quartile analysis is insightful, the rationale for the specific quartile cutoffs could be better explained.

5. The interaction effects in subgroup analyses were explored, but the manuscript lacks a robust discussion of these interactions' clinical or biological significance.

6. It is quite hard to follow the table's presentation. The tables are informative but overwhelming because all data are in table forms. Adding figures (e.g. plots, etc.) to represent the relationship between UHR and depressive symptoms visually could enhance the understanding of key findings.

7. Overgeneralization that UHR could serve as a "potential biomarker for assessing depression risk" might be premature without further validation in diverse populations or experimental studies (this is not well captured in the limitation of the study).

8. In the discussion, while some plausible mechanisms are discussed, they rely heavily on secondary sources. Further emphasis on how UHR directly interacts with pathways involved in depression would strengthen the argument.

9. Certain manuscript parts have redundant statements, particularly in the discussion. These could be streamlined to improve readability. The introduction requires further refinement to enhance the clarity and focus of the study.

10. While the study explores a novel association, the justification for using UHR over other established metabolic markers for depression risk is not convincingly argued. Expanding on why UHR is superior or complementary would enhance its scientific contribution.

Reviewer #2: The title could be improved by changing relationship into association as its the main topic of investigation.

Abstract needs concluding statement for proper justification.

The rationale for choosing UHR over traditional metabolic syndrome markers could be expanded.

The study uses cross-sectional data but does not sufficiently acknowledge its limitation in establishing causation.

Explicitly mention that causation cannot be inferred and suggest longitudinal studies for future research.

The use of a CESD-10 cutoff of ≥10 to define depression should be referenced with relevant studies which is not enough with the provided references.

It is unclear whether all potential confounders were adequately controlled in the regression models.

The study mentions interaction tests but does not explain how they were performed or interpreted.

6. PLOS authors have the option to publish the peer review history of their article (what does this mean? ). If published, this will include your full peer review and any attached files.

**Do you want your identity to be public for this peer review?** For information about this choice, including consent withdrawal, please see our Privacy Policy .

Reviewer #1: No

Reviewer #2: No

---

## [Author Response · Author response to Decision Letter 1]

30 Jan 2025

The word version has been uploaded and replied to the reviewer

---

## [Editor Report · Decision Letter 1]

4 Feb 2025

Association between the ratio of serum uric acid to high density lipoprotein cholesterol and depressive symptoms in middle-aged and elderly Chinese

PONE-D-24-56380R1

Dear Dr. xu,

We’re pleased to inform you that your manuscript has been judged scientifically suitable for publication and will be formally accepted for publication once it meets all outstanding technical requirements.

Kind regards,

Vinod Kumar Yata, PhD

Academic Editor

PLOS ONE
---

## [Editor Report · Acceptance letter]

PONE-D-24-56380R1

PLOS ONE

Dear Dr. xu,

I'm pleased to inform you that your manuscript has been deemed suitable for publication in PLOS ONE. Congratulations! Your manuscript is now being handed over to our production team.

Kind regards,

on behalf of

Dr. Vinod Kumar Yata

Academic Editor

PLOS ONE